# Structure and Photoluminescence Properties of Thermally Synthesized V_2_O_5_ and Al-Doped V_2_O_5_ Nanostructures

**DOI:** 10.3390/ma14020359

**Published:** 2021-01-13

**Authors:** Chih-Chiang Wang, Chia-Lun Lu, Fuh-Sheng Shieu, Han C. Shih

**Affiliations:** 1Department of Materials Science and Engineering, National Chung Hsing University, Taichung 40227, Taiwan; wilbur0913@gmail.com; 2Department of Chemical Engineering and Materials Science, Chinese Culture University, Taipei 11114, Taiwan; ken.lu1988@gmail.com

**Keywords:** Al-doped V_2_O_5_ nanostructures, photoluminescence, vapor-solid mechanism

## Abstract

Al-free and Al-doped V_2_O_5_ nanostructures were synthesized by a thermal-chemical vapor deposition (CVD) process on Si(100) at 850 °C under 1.2 × 10^−1^ Torr via a vapor-solid (V-S) mechanism. X-ray diffraction (XRD), Raman, and high-resolution transmission electron microscopy (HRTEM) confirmed a typical orthorhombic V_2_O_5_ with the growth direction along [110]-direction of both nanostructures. Metallic Al, rather than Al^3+^-ion, was detected by X-ray photoelectron spectroscopy (XPS), affected the V_2_O_5_ crystallinity. The photoluminescence intensity of V_2_O_5_ nanostructure at 1.77 and 1.94 eV decreased with the increasing Al-dopant by about 61.6% and 59.9%, attributing to the metallic Al intercalated between the V_2_O_5_-layers and/or filled in the oxygen vacancies, which behaved as electron sinks. Thus the Al-doped V_2_O_5_ nanostructure shows the potential applications in smart windows and the electrodic material in a Li-ion battery.

## 1. Introduction

Vanadium is a multi-valent element with oxidation states of V^2+^, V^3+^, V^4+^, and V^5+^, and therefore has several phase-states, including vanadium monoxide (VO), vanadium sesquioxide (V_2_O_3_), vanadium dioxide (VO_2_), and vanadium pentoxide (V_2_O_5_) [1,2]. Vanadium pentoxide (V_2_O_5_) is the most stable phase; it has an orthorhombic structure and layered VO_5_ structures of square pyramids that share corners and edges [3,4]. The layered structure results in the V3d split-off conduction band so that V_2_O_5_ has direct and indirect band gaps at 2.3 and 1.9 eV [5]. Oxygen vacancies ($V_{Ö}$) are the most common defects in V_2_O_5_, and especially in the oxygen layer in the layered VO5 structure [6]. The conduction band that has split off and the $V_{Ö}$-defects provide a flexible pathway for luminescence. Doping with elements and the formation of the core-shell nanostructures modify the photoluminescence properties [7,8]. Many works have reported upon methods of fabricating V_2_O_5_, which are: atomic layer chemical vapor deposition [9], the sol-gel process [10], thermal deposition [11], the hydrothermal process [12], and the thermal evaporation process [7,8]. The photoluminescence property of V_2_O_5_ can be modified by the different dopants, such as Er [13], Co [14], Nd [15,16], Gd [16]; Al [10], and Ga [7]. The above doped V_2_O_5_ materials possess potential applications in light emitters, photocatalysts, and cathodic materials for ionic batteries. Owing to the layered structures and tunable photoluminescence properties, V_2_O_5_ has many applications, including electrochromic devices [17], solar cells [18,19], catalysts [20], solid-state batteries [21], gas sensors [22], chemical species sensors [23], smart windows [4], etc. The dopants as mentioned above were not metallic powders. Instead, the nitrate [10,13,15], sulfate [14], and oxide powders [16] were the dopants, which implied that the metallic ions were easy to form during the synthesized process, resulting in the subsequent influence in photoluminescence. However, the zero-valent metallic ion can also show the influence in the photoluminescence property. Compared to previous literature [7,10,13,14,15,16], this work shows for the first time the metallic Al powders acting as dopants and doping V_2_O_5_ nanostructures. This implies that the zero-valent and/or tri-valent Al ions can be observed, and their influence in the photoluminescence property of the V_2_O_5_ nanostructures can be studied. Al-doped V_2_O_5_ shows potential applications including as photodiode [24], in electrocatalysis [25], thin-film batteries [26], and as cathodic materials for lithium ion batteries [10].

In this work, metallic Al powder was directly used as a source of metallic Al dopant. Both Al-free and Al-doped V_2_O_5_ nanostructures (NSs) were fabricated on Si(100) substrate using a thermal-chemical vapor deposition (CVD) process via a catalyst-free vapor-solid (V-S) mechanism at 850°C under 1.2 × 10^−1^ Torr. In addition, Ar gas was introduced during the whole synthesized process to lower the oxygen partial pressure and to prevent the formation of the Al_2_O_3_ phase. The effects of Al doping on the crystal structures, binding energies, bonding vibration modes, and luminescence properties of V_2_O_5_ NSs are systematically analyzed based on the results of X-ray diffraction (XRD), high-resolution transmission electron microscopy (HRTEM), and Raman, and photoluminescence spectra.

## 2. Materials and Methods

### 2.1. Fabrication of Al-Free and Al-Doped V_2_O_5_ Nanostructures

Al-free V_2_O_5_ NSs were deposited on the Si (100) substrates by the catalyst-free V-S mechanism using the thermal-CVD process at 850 °C in a quartz tube furnace. A 0.3 g mass of high-purity V_2_O_5_ powder (Aldrich, Shanghai, China, 99.5%) was placed in an alumina crucible in the center of the quartz tube; about 10cm downstream of the crucible was placed the Si (100) substrate. The system pressure was pumped down to 4.0 × 10^−2^ Torr and the temperature was increased to 850 °C at 25 °C/min in Ar 30 standard cubic centimeter per minute (sccm) V_2_O_5_ powder was evaporated at 850 °C for 1.5 h, and its vapor was mixed with Ar (30 sccm) and O_2_ (20 sccm) at 1.2 × 10^−1^ Torr; V_2_O_5_ NSs were grown on Si(100) substrate and then cooled to room-temperature at 30 sccm Ar. The final products were denoted as AV0 NSs. Al-doped V_2_O_5_ NSs were synthesized in a similar manner. The starting materials were a mixture of 0.3 g powdered V_2_O_5_ with various amounts of powdered Al (Aldrich, 99.9%)—1.5 mg (0.5 wt.%), 3 mg (1 wt.%), and 4.5 mg (1.5 wt.%)—that were co-evaporated at 850 °C to yield AV05, AV10, and AV15, respectively.

Before analyzing processes, all samples were preserved in a glass desiccator under the pressure of 1 Torr. The samples of the AV nanostructures being covered on the Si(100) substrates were directly analyzed using XRD, X-ray photoelectron spectroscopy (XPS), field-emission scanning electron microscopy (FESEM), Raman, and photoluminescence (PL). The AV nanostructures were detached from the Si(100) substrate by ultrasonication in the ethanol solution, followed by the HRTEM observation.

### 2.2. Characterizations

The crystal structures of the AV NSs were determined using a mass absorption coefficient glancing incident X-ray diffractometer with Cu kα radiation source (λ = 0.154 nm, 40 kV, 30 A, Bruker D2 PHASER) in the recording range from 20 to 60° and high-resolution transmission electron microscope with the electron accelerating voltage of 200 kV (HRTEM, JEOL, JEM-3000F, Tokyo, Japan). The chemical binding energies were measured by X-ray photoelectron spectroscope using the Al kα radiation with the electron energy of 1486.6 eV (XPS, Perkin-Elmer model PHI1600 system, Waltham, MA, USA). The chemical bonding vibration modes were identified by Raman spectrometer (3D Nanometer Scale Raman PL microspectrometer, Tokyo Instruments, INC., Tokyo, Japan) with an excitation source of a semiconductor laser at the wavelength of 532 nm. Room-temperature PL spectra were recorded by a confocal Raman spectrometer (Alpha300, Witec, Ulm, Germany) with the source of semiconductor laser (λ = 532 nm, 0.5 mW).

## 3. Results

### 3.1. X-ray Photoelectron Spectroscopy (XPS) Analysis

Figure 1a presents the XPS spectra of the V 2p and O 1s orbitals of AV0 and AV15 NSs. The peaks at 516.4 eV (V 2p^3/2^) and 523.7 eV (V 2p^1/2^) of the AV0 NSs with a splitting Δ-value of 7.3 eV are consistent with the characteristic of orthorhombic V_2_O_5_ structures [27]. The AV15 NSs show a similar splitting Δ-value (7.4 eV), indicating that the Al dopants do not affect the V_2_O_5_ orthorhombic structure. The O 1s peak at 529.3 eV is attributed to the oxygen ions (O_L_) associated with the V–O bonds in the V_2_O_5_ structures [28,29]; the peak at 530.7 eV (O_C_) is attributed to the chemisorbed oxygen on the surface of the NSs [7]. The energy difference between V 2p^3/2^ and O_L_ is about 12.9 eV, indicating the formation of the V_2_O_5_ phase [29]. The values of V 2p and O_L_ in the AV15 NSs (Figure 1a) are similar to those of the AV0 NSs. However, the binding energy of O_C_ in AV15 NSs shifts upward to 532.2 eV relative to that of AV0 NSs (530.7 eV). The reduction potential of Al is as low as −0.677 V [30], indicating that Al more easily reacts with oxygen-containing species, such as oxygen and/or hydroxyl, and forms the amorphous phases of Al–O and/or Al–OH [31] at the surface of the NSs.

Therefore, the O_C_ position in AV15 NSs ends up with a blue-shift. Figure 1b shows that the energies of the Al 2p orbitals in AV05, AV10, and AV15 NSs have the same binding energy of 72.12 eV [32], showing that the Al dopants behave as a metal rather than the ions in the V_2_O_5_ NSs. Table 1 presents the atomic percentage (at.%) of Al in the V_2_O_5_ NSs, indicating that the at.% of Al dopant increases with Al contents.

### 3.2. X-ray Diffraction (XRD) Patterns

The powder XRD analysis of both Al-free and Al-doped V_2_O_5_ NSs reveal the typical orthorhombic V_2_O_5_ structure (joint committee on powder diffraction standard (JCPDS) 77-2418) with the planes (001), (110), (400), (111), (002), and (600) at diffraction angles of 20.36, 26.16, 31.06, 33.01, 41.37, and 47.35°, correspondingly, as presented in Figure 2. No irrelevant phases, such as Al_2_O_3_ and metallic Al, were detected. The insets in Figure 2a reveal that the full width at half maximum (FWHM) of the V_2_O_5_ (001) plane varies with the Al content and show a deviation of ±2.9% of all the AV NSs, indicating that the Al dopants do not affect the V_2_O_5_ crystal structure.

Based on the Bragg law and the plane-spacing equation for an orthorhombic structure, $\frac{1}{d_{hkl}^{2}}=\;\frac{h^{2}}{a^{2}}+\frac{k^{2}}{b^{2}}+\frac{l^{2}}{c^{2}}$, the lattice constants (a, b, and c), and the ratios c/a and c/b can be estimated; they are provided in Table 2. Both c/a and c/b ratios increase with the addition of Al dopant, as shown in Figure 2b. The V_2_O_5_ comprises packed layered [VO_5_]–[VO_5_] structures, facilitating the formation of $V_{Ö}$ between the layers [8,11]. The radius of the Al^0^ is 1.43 Å similar to the $V_{Ö}$(1.4 Å). The interlayer spacing of the layered V_2_O_5_ structure is about 4.32 Å [33].

Therefore, the increase in c/a and c/b can be attributed to the filling by Al^0^ of the sites of $V_{Ö}$ and/or intercalating between the two layered VO_5_ structures in V_2_O_5_ NSs. The Al dopant slightly increases the lattice distance between the two [VO_5_]-layers in the a–c plane (inset in Figure 2b [34]) by approximately 0.1–0.2% when the Al content is ≤1 wt.% and a more significant increase of the lattice distance (by 1.2%) at an Al-dopant content of 1.5 wt.%. The lattice spacing in the a–b plane (inset in Figure 2b [34]) significantly increases with the incorporation of Al dopant. These results reveal that the c/b and c/a ratios increase with the incorporated Al dopants in the V_2_O_5_ NSs.

### 3.3. Field-Emission Scanning Electron Microscopy (FESEM) and High-Resolution Transmission Electron Microscopy (HRTEM) Analysis

FESEM images of AV0 and AV10 NSs are shown in Figure 3a,b. Most of the nanostructures exhibit the nanowire-like shape. Their crystal structure and growth direction are estimated by the following HRTEM analysis and shown in Figure 3c,d. Inset 1 (Figure 3c) shows d-spacings of 5.7 and 3.6 Å, consistent with the planes V_2_O_5_ (100) and V_2_O_5_ (110); the angle between the (100) and (110) planes is 72°, consistent with the selective area diffraction (SAD) pattern in inset 2 (Figure 3c). The d-spacings of AV10 NSs, as shown in inset 3 (Figure 3d), are 4.2 and 3.4 Å, consistent with the planes V_2_O_5_ (001) and V_2_O_5_ (110) and the angle between (001) and (110) is 89.5°, consistent with the SAD pattern in inset 4 (Figure 3d). The theoretical angles between (110) and (200) and between (110) and (001) are 73° and 90°, respectively, which are estimated using the formula, $\phi \;=\;cos^{-1}\left[\frac{\frac{h_{1}h_{2}}{a^{2}}+\frac{k_{1}k_{2}}{b^{2}}+\frac{l_{1}l_{2}}{c^{2}}}{\sqrt{\left(\frac{h_{1}^{2}}{a^{2}}+\frac{k_{1}^{2}}{b^{2}}+\frac{l_{1}^{2}}{c^{2}}\right)\left(\frac{h_{2}^{2}}{a^{2}}+\frac{k_{2}^{2}}{b^{2}}+\frac{l_{2}^{2}}{c^{2}}\right)}}\right]$, where ϕ is the angle between the (h_1_k_1_l_1_) plane and the (h_2_k_2_l_2_) plane; a (11.51 Å), b (3.559 Å), and c (4.371 Å) are the lattice constants of the orthorhombic V_2_O_5_ structure Based on the above discussion, the growth direction of the AV NSs can thus be confirmed as being in the [110]-direction.

### 3.4. Raman Spectra

The V_2_O_5_ structure is orthorhombic and has a *P_mnm_* symmetry; V_2_O_5_ has typical Raman active peaks of AV0 and AV10 NSs which fitted by Gaussian deconvolution are observed at around 141.71, 195.26, 282.04, 302.64, 403.51, 480.45, 524.80, 699.93, and 993.36 cm^−1^ [3,35,36,37], as shown Figure 4a and listed in Table 3. The crystal structure of V_2_O_5_ NSs is layered V_2_O_5_s that is packed along the c-axis of the unit cell, and each V_2_O_5_ layer comprises square pyramids of VO_5_ that share edges and corners, as shown in Figure 4b [8,38]. One vanadium atom is connected to six oxygen atoms with five bond lengths, which are of 1.557, 1.779, 1.878, 2.017, and 2.791 Å, as shown schematically in Figure 4b [38]. The V–O1 bond has the strongest binding energy but the shortest bonding length (1.557 Å) of all of the V–O bonds; the V–O1′ bond is a weak Van der Waals’ bond between layered-VO_5_ with a bonding length of 2.791 Å; along the b-axis, the V–O21 and V–O22 bonds have a length of 1.878 Å; along the a-axis, the V–O bonds are of two types:(1)The V–O2′1 bond has a length of 2.017 Å, and(2)The V–O3 bond has a length of 1.779 Å.

**Figure 4 materials-14-00359-f004:**
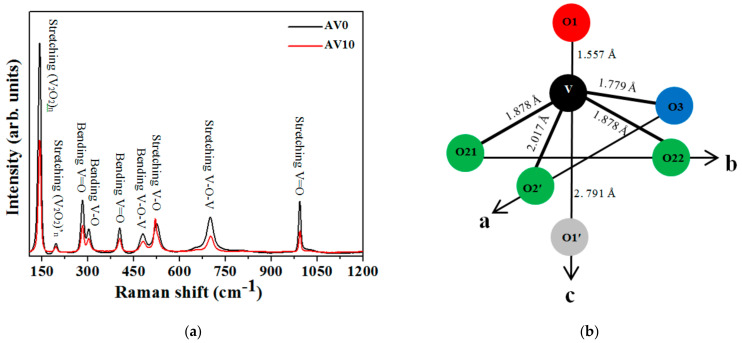
(**a**) Raman spectra of the AV0 and AV10 NSs, and (**b**) schematic V_2_O_5_ molecular Scheme [38].

**Table 3 materials-14-00359-t003:** Characteristic Raman shifts of the nanostructures of AV0 and AV10 NSs.

Raman Scattering Modes	AV0 NSs		AV10 NSs		Deviation of the Raman Shifts (%)
	Raman Shifts (cm^−1^)	Intensity	Raman Shifts (cm^−1^)	Intensity	
Stretching (V_2_O_2_)_n_	141.71	84,745.82	141.93	45,097.12	0.071
Stretching (V_2_O_2_)’_n_	195.26	3371.40	194.45	1830.32	0.051
Bending V=O	282.04	20,894.35	282.14	10,431.08	0.017
Bending V–O	302.64	9033.58	302.47	5007.98	0.016
Bending V=O	403.51	9475.78	403.33	4967.71	0.025
Bending V–O–V	480.45	6944.95	481.35	3846.47	0.062
Stretching V–O	524.80	11,097.60	522.07	11,202.89	0.258
Stretching V–O–V	699.93	12,863.33	701.25	5757.458	0.099
Stretching V=O	993.36	20,228.08	994.02	8188.87	0.030

This orthorhombic V_2_O_5_ structure has two bonding vibrations. They are: (1) stretching mode: the terminal oxygen of the V=O bond is the unshared oxygen (O1) and yields a peak at 993.36 cm^−1^ [39,40]; the doubly coordinated oxygen in the V–O–V bond is a corner-sharing oxygen in one of the two adjacent pyramids yields a peak at 699.93 cm^−1^ [41]; the triply coordinated oxygen of the V–O bond is an edged-sharing oxygen in three adjacent pyramids yields a peak at 524.80 cm^−1^ [6,39]; and (2) bending mode: the bridging V–O–V bond at 480.45 cm^−1^ [39,41]; the terminal oxygen associated with the V=O bond yields a peak at 403.51 cm^−1^ [40]; the triply coordinated oxygen associated with the V–O bond yields a peak at 302.64 cm^−1^ [6]; the V=O terminal oxygen associated with the bond yields a peak at 282.04 cm^−1^; (V_2_O_2_)_n_ and (V_2_O_2_)’_n_ bonds yield peaks at 194.26 and 141.71 cm^−1^, respectively, corresponding to the chain translation and strongly associated with the layered structures [41]. Table 3 presents the Raman shifts of the AV0 and AV10 NSs. The deviations of the Raman shifts upon the addition of Al reveal a slight difference between AV0 and AV10 NSs, indicating that Al has no major effect on the V_2_O_5_ binding structures, which is consistent with the XPS results.

The intensity of the Raman peaks (as shown in Figure 4a and Table 3) clearly decreases upon the addition of Al dopant. Jung et al. [42] reported a decrease in the Raman intensity from V_2_O_5_ upon the insertion of Li into the Li_x_V_2_O_5_ lattice owing to the formation of negative charge carriers by the reduction of V^5+^ to V^4+^ [43]; Park et al. determined that the decreasing Raman intensity is attributable to the extraction of Li from the Li_x_CoO_2_ lattice because the oxidization state increases from Co^3+^ to Co^4+^, generating positive charge carriers [44,45]. Both of these cases are attributable to the metallization of the host materials, so decreases in Raman intensity arise from the reduction of the optical skin depth of the Raman excitation light [42]. In this work, metallic Al is doped into the V_2_O_5_ lattice, as revealed by XPS and XRD; the host materials, therefore, exhibit more metallic characters. Hence, the Raman intensity of the V_2_O_5_ NSs clearly decreases with the addition of Al (Figure 4a).

### 3.5. Photoluminescence (PL) Spectra

Figure 5a presents the PL spectra of the AV NSs, which reveal two prominent emissions at 1.77 (E1) and 1.94 eV (E2). V_2_O_5_ has split-off conduction bands owing to its layered structure, suggesting that it has several energy bands from the top of the O2p valance band to the split-off V3d conduction band [11]. The V_2_O_5_ crystal structure comprises from the [VO_5_]-[VO_5_] layers and oxygen vacancies ($V_{Ö}$) easily being formed between the layers [11]. Accordingly, the $V_{Ö}$-related emission E1 occurs at 1.77 eV and E2 at 1.94 eV is attributed to the intrinsic emission band, which is split off from the V3d conduction band in the V_2_O_5_ NSs [6,46]. Figure 5b presents variations of the intensities of E1 and E2 with the Al content. Such variations in luminescence intensities decay exponentially as Al content increases. The intensity of E1 decreases by about 61.6% and that of E2 decreases by about 59.9% as the Al content increases from 0 to 1.5 wt.%.

Table 4 shows the influence in the photoluminescence intensity of the V_2_O_5_ by different dopants, including metallic Al, metallic Ga [7], aluminum nitrate nonahydrate (Al(NO_3_)_3_·9H_2_O) [10], erbium nitrate pentahydrate (Er(NO_3_)_3_·5H_2_O) [13], cobalt sulfate heptahydrate (CoSO_4_·7H_2_O) [14], gadolinium oxide (Gd_2_O_3_) [16], neodymium oxide (Nd_2_O_3_) [16], aluminum nitrate (Al(NO_3_)_3_) [24], and aluminium oxide (Al_2_O_3_) [26]. Besides the Ga dopants, the PL intensities of V_2_O_5_ decrease with the zero-valent Al the same as the results of the tri-valent elements.

### 3.6. Proposed Luminescence Mechanism

Figure 6 presents a possible mechanism for E1 and E2 from the AV NSs. The conduction band (E_C_), Fermi level (E_F_), and valence band (E_V_) of the V_2_O_5_ structures in the Al-free V_2_O_5_ NSs, are at −4.7, −5.45, and −7.00 eV [47], respectively, as shown in Figure 6a. Some of the excited electrons in the E1 level were generated by an incident light from the E_v_ of the V_2_O_5_ transiting to the E2 level, causing the intensity of E2 to exceed that of E1, as shown in Figure 5b. According to XPS and XRD results, the metallic Al exists in the V_2_O_5_ NSs, which can be located between the layered V_2_O_5_ structure and/or the oxygen vacancy. The Fermi level of the metallic Al is approximately 4.08–4.28 eV [48]. Hence, in the Al-doped V_2_O_5_ NSs, as shown in Figure 6b, these excited electrons in the E2 level, which were excited by the incident radiation, can transit along three pathways. These pathways are (1) from E2 to E1, and then to the Fermi level of the metallic Al; (2) from E2 to the work-function level of the metallic Al; and (3) from E2 to E1. Along pathway 1, the excited electrons transit to the metallic Al, so the E2 intensity decreases; along pathway 2, the electrons that are generated upon the formation of the $V_{Ö}$-defects transit to the metallic Al, leading the decreasing E1 intensity; along pathway 3, the excited electrons in the E2 level transit to the E1 level, so the intensity of E2 is lower than that of E1 for all of the AV NSs. Moreover, the concentration of $V_{Ö}$ is reduced upon the addition of Al dopant because the sites of the $V_{Ö}$ defects are filled by Al^0^. Then the intensity of E1 emission thus tends to decrease. These results suggest that Al as the dopant plays the following three roles in the AV NSs system: (1) as an electron sink, (2) filling the sites of the $V_{Ö}$ defects and reducing the $V_{Ö}$ concentration, and (3) intercalating the layered V_2_O_5_ structure. Therefore, the intensity of E1 and E2 both decreases as the Al dopant concentration increases.

## 4. Conclusions

Both Al-free and Al-doped V_2_O_5_ NSs were fabricated on the Si (100) substrate by a thermally activated CVD process at 850 °C via a V-S mechanism. XPS revealed the V–O binding energy of the V_2_O_5_ phase and the presence of metallic Al rather than Al^3+^ ions. XRD revealed the typical orthorhombic phase of the V_2_O_5_ with c/a and c/b ratios that increase with the Al dopant concentration, suggesting that Al^0^ (radius = 1.43 Å) preferentially fills the $V_{Ö}$ (radius = 1.4 Å) sites owing to their similar sizes. The FWHM has a similar value with a deviation of ±2.9% upon the addition of Al dopant, indicating that Al does not affect the V_2_O_5_ crystal structure. HRTEM verified that the growth direction of both Al-free and Al-doped V_2_O_5_ NSs is in the [110]-direction. Raman shifts revealed slight deviations upon the addition of Al, indicating that the Al content does not affect the binding structure of the V_2_O_5_. The photoluminescence results proved that the emission intensities at 1.77 and 1.94 eV decreased by factors of 61.6% and 59.9%, respectively, as the concentration of Al dopant increases from 0 to 1.5 wt.%. The decrease in the luminescence intensities was attributable to the following causes: (1) Al dopants reduced the $V_{Ö}$-defect concentration and (2) Al acted as an electron sink. Therefore, the recombination rate of electrons and holes decreases, reducing the luminescence intensities at 1.77 and 1.94 eV, upon doping with Al. Al-doped V_2_O_5_ NSs have potential applications in photocatalysts, solar cells, gas sensors, and electrodic materials.

## Figures and Tables

**Figure 1 materials-14-00359-f001:**
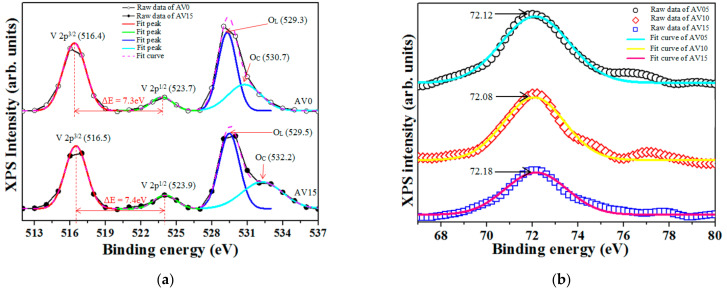
X-ray photoelectron spectroscopy (XPS) spectra of V 2p and O 1s of (**a**) AV0 and AV15 NSs (Al-doped V_2_O_5_ nanostructures). (**b**) Al 2p of AV NSs.

**Figure 2 materials-14-00359-f002:**
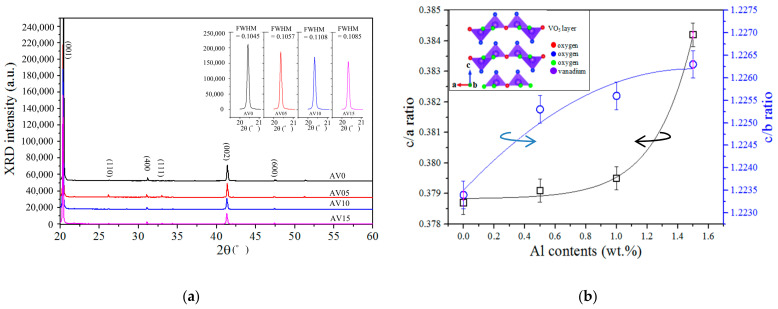
(**a**) Powder X-ray diffraction (XRD) analysis of AV NSs and the inset showing FWHMs varying Al dopants in V_2_O_5_ (001); (**b**) c/a and c/b ratios varying with Al dopant (wt.%); the inset depicting the V_2_O_5_ molecular structures [34].

**Figure 3 materials-14-00359-f003:**
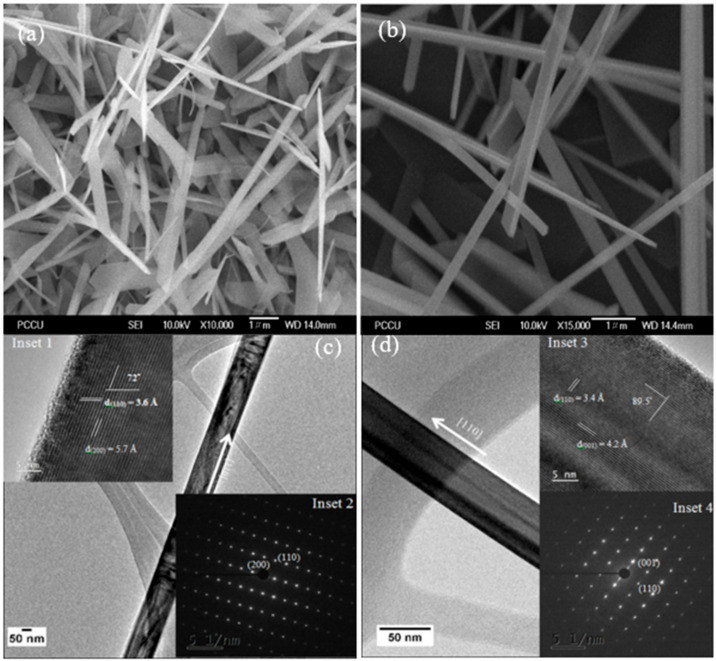
Field-emission scanning electron microscopy (FESEM) images of (**a**) AV0 and (**b**) AV10 nanostructures. High-resolution transmission electron microscopy (HRTEM) images of (**c**) AV0 and (**d**) AV10 NSs, Insets 1 and 3 showing the high resolution images of the AV0 and AV10 NSs, respectively; while inset 2 and 4 showing the SAD pattern of the AV0 and AV10 NSs.

**Figure 5 materials-14-00359-f005:**
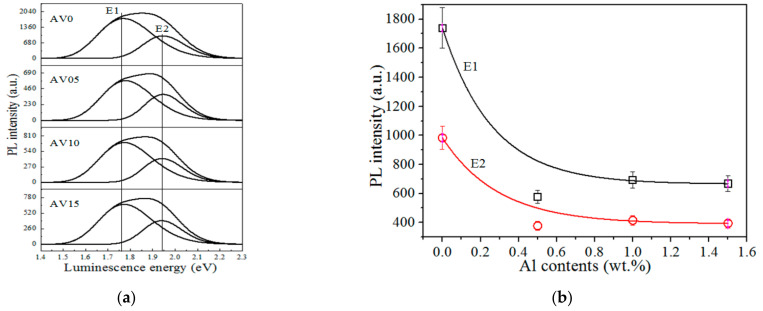
(**a**) Photoluminescence (PL) spectra of the samples AV0, AV05, AV10, and AV15 NSs; and (**b**) intensities of the E1 and E2 varying with the Al contents.

**Figure 6 materials-14-00359-f006:**
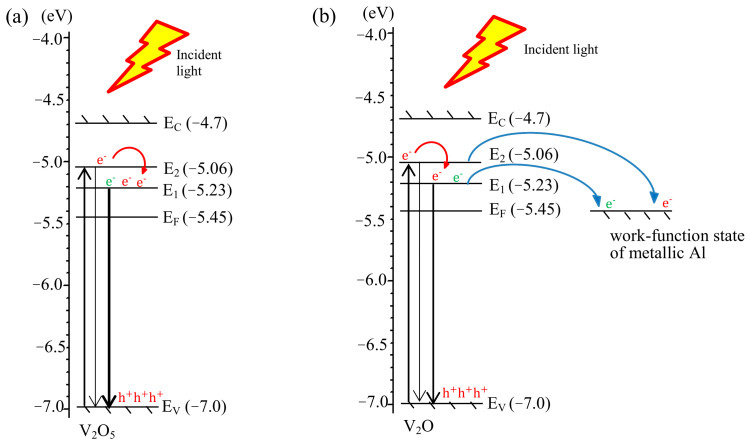
Proposed mechanism of the migration pathways for the electrons and holes in (**a**) Al-free and (**b**) Al-doped V_2_O_5_ NSs at an equilibrium state.

**Table 1 materials-14-00359-t001:** Al at.% of the AV NSs with various Al contents (wt.%).

Samples	AV0	AV05	AV10	AV15
Al contents(g)	0	0.0015	0.003	0.0045
Al contents (wt.%)	0	0.5	1	1.5
Al contents (at.%)	0	4.45	8.64	11.31

**Table 2 materials-14-00359-t002:** Lattice constants, c/a, and c/b of the AV NSs with various Al contents.

Al Contents (wt.%)	Diffraction Angle (2θ)			Lattice Constants of V_2_O_5_ (Å)			c/a	c/b
	V_2_O_5_ (001)	V_2_O_5_ (110)	V_2_O_5_(400)	a	b	c		
0	20.36	26.17	31.07	11.504	3.562	4.357	0.3787	1.2234
0.5	20.42	26.28	31.19	11.461	3.546	4.345	0.3791	1.2253
1.0	20.35	26.19	31.10	11.492	3.558	4.361	0.3795	1.2256
1.5	20.33	26.19	31.09	11.494	3.558	4.363	0.3842	1.2263

**Table 4 materials-14-00359-t004:** The influence in PL intensities of V_2_O_5_ in various dopant precursors.

Dopant Precursors	Dopant Type	PL Intensity of the V_2_O_5_	Ref.
metallic Al	Al^0^	Decreases with the increasing precursor contents	This work
metallic Ga	Ga^3+^	Enhancement at proper precursor contents	[7]
Al(NO_3_)_3_·9H_2_O	Al^3+^	Decreases with the increasing precursor contents	[10]
Er(NO_3_)_3_·5H_2_O	Er^3+^	No data	[13]
CoSO_4_·7H_2_O	Co^2+^	Decreases with the increasing precursor contents	[14]
Gd_2_O_3_	Gd^3+^	Decreases with the adding of precursor	[16]
Nd_2_O_3_	Nd^3+^	Decreases with the adding of precursor	[16]
Al(NO_3_)_3_	Al^3+^	Decreases with the adding of precursor	[24]

## Data Availability

Data is contained within the article.

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
