# Peer review of "Structure and Photoluminescence Properties of Thermally Synthesized V2O5 and Al-Doped V2O5 Nanostructures"

_materials, 2021, doi:10.3390/ma14020359_

Round 1

Reviewer 1 Report

In this study by Wang and colleagues, the authors studied photoluminescence as a function of the structure of V2O5 and Al/V2O5 nanostructures. These materials recently received attention from the scientific community, which makes these results timely and potentially interesting. However, several issues must be addressed first before this paper can be reconsidered for publication in Materials. Please find the suggestions below:
1) The level of English should be considerably improved as there are many errors that make it hard to understand the presented concepts. There are also typos, unexpected spaces, wrong capitalization of letters, etc. which should be eliminated.
2) "Vanadium is a multi-valent element with oxidation states of V2+, V3+, V4+, and V5+ 22 , and therefore has several phase-states, including vanadium monoxide (VO), vanadium sesquioxide (V2O3), vanadium oxide (VO2), andvanadium pentoxide (V2O5) [1,2]" (Lines 22-24) - did you mean vanadium dioxide?
3) The novelty factor is not defined in the introduction section. Please make a more detailed demonstration of the state of the art and specify exactly what was accomplished in this contribution that others did not do before.
4) The contribution lacks proper formatting:
- margins in section 2.2. are not consistent with the rest of the file
- plots (e.g. Fig. 1) are not of the same size and extend beyond the margins
- there are no legends to certain plots, so they cannot be studied.
- some of the captions overflow to the following page while they should be present on the same page as the plots that they describe
- page six is empty (!)
- Fig. 3 is enormous
- both panels in Fig. 6 should be of the same size
I cannot list all of these problems, but there are many more of them. I invite the authors to critically look at this contribution and put it into the appropriate shape.
5) More details should be provided regarding the characterization conditions. Please report on the parameters used for TEM, XPS, Raman, and PL examination. Currently, these steps cannot be reproduced and for a paper to be published it must be reproducible.
6) XPS plots should contain the deconvoluted peaks, the envelope, and raw data
7) Please comment on the reproducibility of the synthesis. It appears that the samples were produced only once for each distinctive parameter set.
8) SEM and TEM images in Fig. 3 should be acquired at the same magnification. Currently, they cannot be compared.
9) Ideally, spectra in Fig. 4a should be normalized to each other to enable comparison between them.
10) How did the authors ensure the same concentration of the samples to be analyzed by PL? At present, the PL intensities in Fig. 5 are offset, but it can as well be the result of different concentrations of the analytes.

Reviewer 2 Report

The manuscript submitted to Materials entitled "Structure and photoluminescence properties of thermally synthesized V2O5 and Al-doped V2O5 nanostructures" by Chih-Chiang Wang et al. presents the preparation and photoluminescence studies of Al-doped V2O5 nanostructures synthesized by a thermal-CVD process.

These comments can be found bellow:  

1) In the introduction the authors state that different dopants can fine tune the photoluminescence properties of V2O4. A few general applications are associated with these doped materials. It would be interesting if the authors could associate the dopant in V2O5 to the most common application. It would make the introduction more clear.  

2) Also, it would be interesting if the authors would add a few lines about V2O4 nano materials doped with aluminium and discuss the major applications where they are used.  

3) The characterisations sections lacks information. The authors should add all the specifications and when possible the way the samples were prepared for all the characterisation present in this manuscript.  

4) Are the XRD patterns Powder DRX analysis? This should be stated in the figure caption. Furthermore, in figure 2a in the x axis the authors should place “degrees” after the 2Theta.  

5) In figure 4a the authors should remove the symbols that overlap the curves. It makes the figure a little bit confusing and it is possible to change the colour of the curves of each peak enhancing the quality of presentation. The attributions are overlapping the curves, and this should also be changed.  

6) The author should add a paragraph comparing the photoluminescence characteristics of these materials with other V2O4 materials already published in the literature. The reason why these materials present themselves as advantageous or not when compared with those materials should also be discussed. This could be added to the conclusion.

In general, the authors did a good job on this research and the manuscript is written logically, with the overall claim well supported by the results. The structure of the article is simple, with a broad discussion of the importance of Al-doped V2O5 nanostructures, and the photoluminescence properties and mechanisms explained.

I don't have concerns about the scientific part and the results of the current study, are interesting. Apart from this, please check the manuscript carefully for typos and wording to make it ready to publish.

Thank you.

Round 2

Reviewer 1 Report

Thank you for responding to the given suggestions. However, the formatting of the article is again not consistent.
- There are considerable differences in interline spacing e.g. compare pages 1 and 2.
- The inset in Fig. 2a should be enlarged as it is not legible.
- Some of the headlines are separated from corresponding sections e.g. page 2.
- There is an unexpected empty space at the beginning of pages 7 and 8.
These issues, however, may be handled at the proofing stage. The article can be accepted for publication upon the incorporation of the above-mentioned corrections.
